# The Efficacy of Trabecular Titanium Cages to Induce Reparative Bone Activity after Lumbar Arthrodesis Studied through the 18f-Naf PET/CT Scan: Observational Clinical In-Vivo Study

**DOI:** 10.3390/diagnostics12102296

**Published:** 2022-09-23

**Authors:** Fabio Cofano, Daniele Armocida, Livia Ruffini, Maura Scarlattei, Giorgio Baldari, Giuseppe Di Perna, Giulia Pilloni, Francesco Zenga, Elena Ballante, Diego Garbossa, Fulvio Tartara

**Affiliations:** 1Neurosurgery Unit, Department of Neuroscience, University of Turin, 10124 Turin, Italy; 2Spine Surgery Unit, Humanitas Gradenigo, 10153 Turin, Italy; 3Neurosurgery Unit, Department of Human Neuroscience, Sapienza Università of Rome, 00185 Rome, Italy; 4Nuclear Medicine Unit, University Hospital of Parma, 43100 Parma, Italy; 5Neurosurgery Unit ASST Fatebenefratelli Sacco, 20121 Milan, Italy; 6BioData Science Center, IRCCS Mondino Foundation, 27100 Pavia, Italy; 7Department of Mathematics, University of Pavia, 27100 Pavia, Italy; 8Neurosurgery Unit, ICCS Città Studi, 20131 Milan, Italy

**Keywords:** lumbar arthrodesis, titanium trabecular cages

## Abstract

*Background:* Titanium trabecular cages (TTCs) are emerging implants designed to achieve immediate and long-term spinal fixation with early osseointegration. However, a clear radiological and clinical demonstration of their efficacy has not yet been obtained. The purpose of this study was to evaluate the reactive bone activity of adjacent plates after insertion of custom-made titanium trabecular cages for the lumbar interbody with positron emission tomography (PET)/computed tomography (CT) 18F sodium fluoride (18F-NaF). *Methods:* This was an observational clinical study that included patients who underwent surgery for degenerative disease with lumbar interbody fusion performed with custom-made TTCs. Data related to the metabolic-reparative reaction following the surgery and its relationship with clinical follow-up from PET/CT performed at different weeks were evaluated. PET/CTs provided reliable data, such as areas showing abnormally high increases in uptake using a volumetric region of interest (VOI) comprising the upper (UP) and lower (DOWN) limits of the cage. *Results:* A total of 15 patients was selected for PET examination. Timing of PET/CTs ranged from one week to a maximum of 100 weeks after surgery. The analysis showed a negative correlation between the variables SUVmaxDOWN/time (r = −0.48, *p* = 0.04), ratio-DOWN/time (r = −0.53, *p* = 0.02), and ratio-MEAN/time (r = −0.5, *p* = 0.03). Shapiro−Wilk normality tests showed significant results for the variables ratio-DOWN (*p* = 0.002), ratio-UP (0.013), and ratio-MEAN (0.002). *Conclusions:* 18F-NaF PET/CT has proven to be a reliable tool for investigating the metabolic-reparative reaction following implantation of TTCs, demonstrating radiologically how this type of cage can induce reparative osteoblastic activity at the level of the vertebral endplate surface. This study further confirms how electron-beam melting (EBM)-molded titanium trabecular cages represent a promising material for reducing hardware complication rates and promoting fusion.

## 1. Introduction

Vertebral fusion is the most common and effective intervention for the treatment of lumbar disc degeneration [1], and the introduction of minimally-invasive techniques has gained popularity, lowering surgery-related traumatic insult [2,3]. However, the occurrence of failed-back-surgery syndrome after arthrodesis still remains a problem in 5% to 15% of lumbar-spine-fusion cases [4]. To reduce the rate of failure, scientists have, in recent decades, moved to investigating new concepts concerning techniques and instrumentation. Further, the evaluation of the sagittal balance and pelvic parameters has assumed increasing value becoming an element of analysis and discussion. High pelvic incidence-lumbar-lordosis mismatch has been shown to be associated with adjacent segment degeneration, increased risk of revision surgery, and residual or worsened postoperative symptoms [5,6,7]. For these reasons, the restoration and long-term preservation of segment lordosis (SL) is now considered one of the major goals of vertebral-fusion procedures. The use of interbody fusion with intersomatic cages is the most widely used and effective method to increase and correct SL [8]. Restoration of intervertebral-disc lordosis and height with iperlordotic-cage insertion produces a significant axial load [9] on the vertebral endplate, enhancing sagittal correction [8].

The most frequent complication of interbody fusion is the cage subsidence within vertebral endplates with consequent reduction of disc height and loss of segmental lordotic correction [9,10]. This also results in a reduced mechanical stability leading to pseudoarthrosis and poorer clinical outcomes. Time of subsidence has been reported as varying from some weeks to several months and many factors can contribute to its occurrence, including bone quality, cage morphology, and materials [11,12,13]. Therefore, the need to generate stable lordotic corrections has also brought attention to the use of materials able to guarantee primary stability and rapid biological-integration capacity. In this regard trabecular metal cages present remarkable aspects of interest [14].

Trabecular titanium is emerging as a valuable candidate to achieve immediate and long-term spinal correction and stability through effective early osteo-integration.

Recent in vitro studies [15,16] demonstrated that trabecular titanium is a biocompatible surface able to induce mesenchymal stem cell adhesion, proliferation, and differentiation towards the osteogenic lineage, even without the addition of osteogenic factors [17]. Trabecular titanium had a sufficient roughness to allow cell attachment, migration, and proliferation suggesting interesting biological behaviors of the scaffold. It appears very likely that the highly porous contact area between the cage and the vertebral endplates assures an optimal ground for bone ingrowth [18,19].

However, in vivo clinical studies directly demonstrating the effectiveness of the osteo-integrative activity of these materials and their role in preventing subsidence are still lacking. Fluorine-18-sodium fluoride (F-NaF) PET/CT is a relatively new high-resolution bone imaging technique that measures turnover. 18F-NaF PET has previously been shown to correlate with histomorphometric parameters [20] and the osteo-integration capacity of different materials and their use has been increasingly used to accurately assess the images and provide awareness of the normal distribution and major artifacts in imaging [21]. This study aimed to evaluate reactive adjacent endplate bony activity after trabecular titanium cage (TTC) placement in order to assess, in vivo, the potential for implant osteo-integration through the analysis of data extracted from positron-emission tomography/computed tomography (PET/CT) obtained at varying times after surgery using 18F-sodium fluoride (18F-NaF), a tracer chemically absorbed into hydroxyapatite in the bone matrix by osteoblasts able to reflect bone re-modeling.

## 2. Materials and Methods

This was an observational study analyzing prospectively collected data of consecutive patients undergoing single- or multi-level lumbar interbody fusion surgery using TTC for degenerative diseases of the lumbar spine who underwent additional imaging examination with 18F-NaF PET/CT for diagnostic evaluation in cases of persistent/recurrent low back pain to investigate the reactive bone activity of the vertebral endplates adjacent to the implanted cage, where standard imaging (MRI or CT) was inconclusive and showed no common signs of cage failure. Patients were recruited over a period from March 2017 to June 2019. The study was then approved by the ethics committee at our institution (Numb. 15092, 8 April 2019).

Due to the cost of the diagnostic procedure, recruitment stopped when the maximum number of patients approved by the committee was reached. Written informed consent was obtained from each patient prior to their enrolment in the study.

### 2.1. Inclusion and Exclusion Criteria

All consecutive patients scheduled for lumbar-spinal-fixation surgery were eligible for this study.

Patients were included in the study if they underwent a lumbar-spine stabilization procedure in the absence of comorbidities and bone disease. Patients were included if, in the postoperative period, they could undergo and sustain an appropriated follow-up program. Patients evaluated had each had implanted the same type of custom-made TTC: TTCs modeled by CAD/CAM technology by means of electron-beam melting (EBM) technology (MT Ortho s.r.l., Aci Sant’Antonio, CT, Italy) for lumbar interbody fusion performed through a lateral transpsoas (XLIF) and/or anterior retroperitoneal (ALIF) approach. Other inclusion criteria were: a previous surgery for degenerative disease with lumbar or lumbosacral interbody fusion performed with TTCs during the investigation period, spontaneous acceptance and informed consent for execution of 18F-NaF PET/CT, absence of comorbidities that contraindicate the execution of the examination, and full availability to achieve at least one year of follow-up.

Exclusion criteria were: (1) patients aged below 16 years; (2) those who were undergoing scoliosis surgery; (3) patients who had a previous surgical history entailing laminectomy or the application of osteosynthesis material at the target levels; (4) patients with incomplete or incorrect data on clinical, radiological, surgical, or follow-up records; (5) patients with ongoing infectious, traumatic, or oncological spinal diseases; and (6) patients with diagnosed osteoporosis (a T-score less than −2.5 on dual-energy X-ray absorptiometry bone densitometry measurements).

All patients underwent a general medical, neurological, and oncologic evaluation at admission. For all the included patients, we recorded patient-related variables such as sex, age, timing of surgery and PET/CT scan, type of surgery, clinical status before and after surgery with the Oswestry Disability Index (ODI) and NRS score for back pain with at least one year follow-up, and PET/CT data (see below the imaging protocol paragraph). Each patient underwent only one PET examination to limit exposure to radioisotopes.

18F-NaF PET/CT was used in the enrolled patients for clinical assessment in cases of persistent/recurrent back pain to investigate reactive bony activity of the vertebral endplates adjacent to implanted porous titanium cage, when standard imaging (MRI or CT) was inconclusive and did not show common signs of cage subsidence.

### 2.2. Interbody Implants and Surgery

All patients underwent interbody fusion with TTCs via the lateral transpsoas or anterior retroperitoneal approach. After radical discectomy and careful preparation of the vertebral endplates, intervertebral cages (MT Ortho s.r.l., Aci Sant’Antonio, CT, Italy) (Figure 1), were inserted into the disc space. The upper and lower vertebrae were fixed using transpedicular screws connected to titanium rods for primary stabilization. The choice of posterior instrumentation with percutaneous or open technique was made according to the need for open decompression or in degenerative, multilevel coronal misalignment. No anterior plates were used with anterior approaches. The specific design features, such as anterior height, individual lordosis correction angle, and footprint are provided by the medical prescriber; moreover, each device is designed exclusively for specific patient since the upper and lower cage surfaces match perfectly with the respective endplate morphology. The shape of the cage was designed in order to restore local lordosis and to respect the harmony of individual spino-pelvic parameters and the pelvic incidence, pelvic tilt, and lumbar lordosis [20,21]. Neuromonitoring was always used during lateral approaches [22]. All interventions were performed by a single surgeon (F.T.) with extensive experience in anterior and lateral approaches.

### 2.3. 18F-NaF PET/CT Imaging Protocol

The preoperative 18F-NaF PET/CT scan was performed in accordance to the European Association of Nuclear Medicine (EANM) guidelines for bone imaging [23] and the Society of Nuclear Medicine (SNM) practice guidelines for 18F-NaF PET/CT bone scan 1.0. [24] Patients were well hydrated before the study and during the uptake time to enhance renal excretion, reducing radiation exposure. Metal objects were removed to prevent attenuation artifacts. PET/CT images were acquired on a 3D integrated PET/CT system Discovery IQ (GE Healthcare, Milwaukee, WI, USA).

Dynamic PET images of the target vertebral region were acquired in list mode for 30 min (frames/time 6 × 5 s, 3 × 10 s, 9 × 60 s, 10 × 120 s) immediately after i.v. injection of 18F-NaF (1.5–3.7 mbq /kg) diluted in 3–5 mL of saline as a 60 s bolus and flushed with saline. Whole-body PET/CT (WB-PET/CT) was acquired in supine position 60 min after tracer injection. Patients voided immediately prior to the scan to reduce bladder activity and were positioned supine on the tomographic bed with arms elevated over the head. WB-PET/CT protocol included a topogram to define the field of view (FOV) from the base of the skull to the pelvis, followed by a low-dose CT scan (120 kV, 140 mA, pitch 1, collimation 16 × 1.25) for attenuation correction and anatomical correlation, and a PET emission scan (3 min per bed position, diagonal FOV 70 cm, 512 × 512 matrix size). Acquired data were reconstructed by Q Clear (GE Healthcare), a Bayesian penalized-likelihood reconstruction algorithm (strength 350). Images were corrected for injected dose, tracer decay, patient body-weight, and attenuation using the low-dose CT scan. Late 18F-NaF PET/CT image of the site of cages was acquired 90 min after tracer injection.

### 2.4. Metabolic Parameters

Imaging review and analysis of attenuation-corrected PET and CT images were performed using an Advantage Workstation 4.4. (GE Healthcare, Milwaukee, WI, USA).

18F-NaF PET/CT scans were analyzed by three nuclear medicine physicians (M.S., L.R., and G.B.) who were blinded to patients’ clinical history and differences between readers were resolved through consensus.

Areas with abnormally high uptake increases—focal, well-circumscribed, and clearly above the background reference area—were identified and contoured using a volumetric region of interest (VOI) encompassing the upper (UP) and lower (DOWN) cage limit (vertebral endplates).

Standardized uptake values (SUVs) were obtained by normalizing the 18F-NaF concentration in the bone region of interest for injected activity and body weight (SUV  = kbq/mL × body weight (kg)/injected activity (mbq)) [25].

The maximum SUV value (SUVmax) as the hottest voxel within the VOI was calculated for the UP and DOWN VOIs (SUVmaxUP and SUVmaxDOWN) [26]. Mean values between SUVmaxUP and SUVmaxDOWN were evaluated and classified as M-SUV.

Uptake of both UP and DOWN VOIs, as well as the mean between them, were normalized (ratio-UP, ratio-DOWN, ratio-MEAN) relatively to 18F-NaF uptake (SUVmax) in the right hip bone, measured within a VOI placed on the normal-appearing bone, encompassing both the cortex and the marrow space, in order to obtain a baseline uptake of the patient and allow a reliable comparison among patients.

### 2.5. Size, Statistics, and a Potential Source of Bias

The study size was given by the selection of the inclusion criteria. As previously stated, we addressed no missing data because incomplete records were an exclusion criterion. The sample was analyzed with SPSS v18 (SPSS Inc., Released 2009, PASW Statistics for Windows, Version 18.0, Chicago, IL, USA) to outline potential correlations between the investigated variables. Comparisons between nominal variables were made with the chi-square test. The threshold of statistical significance was considered as *p* < 0.05.

As a first step only the most caudal level of fusion was analyzed in order to guarantee the independence of observations. The choice to include in this analysis the most caudal level was due to the greatest load borne. The correlation between time and all the considered variables was evaluated by a Pearson correlation analysis (Pearson coefficient and *p*-value of the related test are reported to evaluate the significance). Scatter plots were drawn to visualize the relationship with a regression line. Then all the dataset was analyzed with a mixed-effects model which can handle non-independent observations, and Shapiro−Wilk tests were performed on the variables to satisfy the normality assumption of the models. This model analyzes the influence of the variable time on the target variable, where the variable subject is considered as the random factor. Final models were fitted using the restricted maximum likelihood (REML) method and the general Satterthwaite approximation for the degrees of freedom.

## 3. Results

A total number of 38 patients underwent arthrodesis with EBM TTCs between March 2017 and June 2019. After selection with the previously described criteria, 15 selected cases (7 M, 8 F) underwent lumbar PET/CT with 18F-NaF. Descriptive data are summarized in Table 1. A total of 35 interbody fusions (26 XLIF, 9 ALIF) were performed in these patients. The most frequently treated level was L4-L5 (12/35). In the 35 patients, 18F-NaF PET/CTs were performed within 6 months of the surgical procedure. In four patients PET/CT examination was performed more than one year after surgery, despite radiological evidence of fusion, because of recurrent back pain revealing no complications involving the arthrodesis. No toxicity or adverse reactions to the tracer were reported by patients However, PET/CTs did not reveal any complication involving the arthrodesis. In one case, L3-L5 arthrodesis was followed by L5-S1 elongation two years later; the patient was submitted to PET/CT 11 weeks after L4-L5 surgery and 4 weeks after elongation.

All patients achieved fusion with clinical improvement of their back pain at the last follow-up when compared to the pre-op evaluation, and no subsidence or surgical revisions was reported, preserving the homogeneity of the sample and the interpretation of 18F-NaF PET data.

### Study of Metabolic Results

Mean value of SUVmax was 20 ± 8.9 for the UP-VOI and 21.8 ± 9.6 for the DOWN-VOI.

The mean value of the ratio-UP was 1.4 ± 1.1 and of the ratio-DOWN was 1.6 ± 1.3. Mean follow-up time was 22.2 months (min/max 13/32). (Table 1). Fusion was achieved and registered in all the cases at one-year follow-up. Considering Pearson correlation, the analysis showed a negative correlation between the variables SUVmaxDOWN/time (r = −0.48, *p* = 0.04), ratio-DOWN/time (r = −0.53, *p* = 0.02), and ratio-MEAN/time (r = −0.5, *p* = 0.03). The *p* was not significant for the variables SUVmaxUP/time (*p* = 0.15), M-SUV/time (*p* = 0.07), or ratio-UP/time (*p* = 0.058) (Figure 2).

Shapiro−Wilk normality tests showed significant results considering the variables ratio-DOWN (*p* = 0.002), ratio-UP (*p* = 0.01), and ratio-MEAN (*p* = 0.002). The *p* was not significant considering as variables SUVmaxDOWN (*p* = 0.05), SUVmaxUP (*p* = 0.93), or M-SUV (*p* = 0.33) (Figure 3).

## 4. Discussion

Trabecular titanium was developed to take advantage of its increased porosity and interconnectivity in order to enable greater bone growth and tissue differentiation [27]. The reason for its wide use is due to its biocompatibility, corrosion resistance, low density, and capacity for osteo-integration [28,29,30,31]. Early preclinical studies [15,16,17] on TTCs showed good adhesion and proliferation of mesenchymal cells on the surface of the scaffold [32,33], suggesting an effective cell-material interaction [15,16,17] with a favorable contribution to improved fusion.

In this study the use of 18F-NaF PET/CT provided some adjunctive information about the metabolic-reparative reaction following the implantation of an interbody fusion cage during clinical follow-up; PET/CT supports the evidence regarding the ability of the cage to reach effective early osteo-integration.(Figure 4 and Figure 5) and provide some indications regarding the time required for the bone-remodeling process (the patients underwent PET examinations at different times and this has undoubtedly helped to give value to the final results).

The investigation was performed to obtain metabolic information related to surgical implant with regard to clinical status. Ratio values of upper and lower endplates, as well as mean ratio showed, in both the Pearson correlation and in Shapiro−Wilk normality tests, that an increase of regional blood flow and bone formation activity is registered after surgery for arthrodesis.

According to our results, tracer uptake was already visible in the first week after implantation. This first finding is probably related to the increase in blood flow after preparation of the vertebral endplate and to the consequent bone-cell activation at the contact surface. Therefore the metabolic activity, probably linked to the osteogenic activity, increases progressively until reaching a peak between approximately the 3rd and 4th month after surgery (Figure 2 and Figure 3). This progressive increase in activity can be correlated, as supported by in vitro studies, to the migration, adhesion, and growth of cells from the vertebral bone tissue and to consequent osteoblastic activity with apposition of the bone matrix within the cage pores [34]. In the later stages, the metabolic hyperactivity on the vertebral endplates decreases progressively and tends to disappears around the 10th–12th month after surgery (Figure 2 and Figure 3). This reduction may be the expression of a lower need for bone remodeling caused by sufficient acquired interbody fusion and stability [35,36]. The reparative phase could tend to run out after the formation of bone bridges between the cages and the vertebral plate. The growth of tissue inside the cage can be the basis for a real osteo-integration avoiding delayed subsidence and ensuring stability of the obtained correction.

Few studies have reported the use of 18F-NaF PET for clinical reasons after spinal fusion. Fischer et al. [37] investigated successful incorporation of cervical and lumbar cages after fusion with the use of PET/CT demonstrating that an unsuccessful fusion with residual stress and micro instability was characterized by increased 18F-NaF uptake [37] This is why some studies started reporting associations between tracer uptake and failed implant incorporations [38]. PET hyperactivity was described by Brans et al. [39] in patients with persisting symptoms after lumbar surgery and was connected to concomitant CT findings of subsidence. An increased uptake has also been registered in other studies of cases with screw-loosening with a specificity of 97.4% in patient-based analysis [40]. A relationship with painful pseudarthrosis was suggested by Peters et al. in a retrospective series of 36 patients, because after surgery 18F-NaF activity was reported to be significantly higher in patients with worse functional scores [41]. Abnormal foci of uptake appear to be related to the patient’s source of pain and could be located at various sites involved in arthrodesis (cages, screws, rods, or bone grafts) helping surgeons to choose the best management and strategy if a surgical revision is considered mandatory [42,43].

In this series, implant stability and fusion were finally confirmed by radiological exams (X-rays of CT), helping us to adequately validate our results. As described by previous studies, the presence of pseudarthrosis, subsidence, or implants loosening could result in abnormal uptake, increasing over time and helping to discriminate between asymptomatic and symptomatic cases requiring revision [41,42,43]. According with our preliminary results [36], the use of custom-made TTCs is confirmed to represent a promising approach in order to achieve good stability promoting active osteo-integration and fusion, while minimizing subsidence rates.

We argue that customization for cages and the use of TTCs allows surgeons to take advantage of implant stability by maximizing the contact surface area, and also to plan appropriate SL restoration when necessary.

The innate osteo-inductive capability of the porous titanium scaffold has been suggested from these data. Cells were able to migrate and colonize the inner portions of the cages indicating that trabecular titanium represents a suitable material with appealing properties in terms of biological behavior. In addition, we can clinically confirm that the use of a fully porous device allows for an homogeneous biomechanical distribution of axial load—thus reducing the risks of subsidence of hollow cages filled with bone substitutes—and for the bone matrix to grow into the cage, rapidly stabilizing the implant, confirming what was shown in pre-clinical studies [34,35,36,37].

Zhang et al. [34] verified, at first, the satisfactory biomechanical properties of TTCs compared with standard solid cages in a finite-element analysis and McGilvray et al. also demonstrated a statistically significant range of motion of 3D printed TTCs compared with PEEK or porous PEEK coated with titanium [35].

In conclusion, 18F-NaF PET/CT proved to be a reliable tool to assess incorporation of spinal implants after surgery, providing some adjunctive confirmation about the metabolic-reparative reaction [44] following the implantation of a TTC.

### Limitations and Further Studies

This is the first prospective study of the radiological demonstration of TTCs using PET/TC.

The main limitations of the study were the exiguity of the sample and the variability of the number of treated levels that empowered the clinical burden of the radiological analysis. In order to avoid unethical exposure of control patients to radioisotopes, there was no control group of standard titanium/PEEK cages. The inclusion of such a control group would also not have supported the aim of the study. PET studies were performed at different times after surgery but not at previously well-defined times. The different timings definitely contributed to the value of the statistical analysis and results. The correlation between data obtained with PET and the data obtained from in vitro experiments was largely based on abstract rational associations.

Among future perspectives, other than its already-described role in diagnosing symptomatic subsidence and implant failure or loosening, the use of 18F-NaF PET/CT could be of importance in the correct clinical diagnosis of adjacent-segment disease and especially in identifying symptomatic discs in the context of a multilevel degenerated spine, playing the part of the outdated spinal discography (Figure 5).

## 5. Conclusions

18F-NaF PET/CT proved to be a reliable tool for investigating the metabolic-reparative reaction following the implantation of an interbody fusion cage. Customization of cages and the use of TTCs allows surgeons to take advantage of implant stability by maximizing the contact surface area, and also, when necessary, to plan appropriate SL restoration. 18F-NaF PET/CT demonstrates that TTCs have a good capacity for induction, over time, of reparative osteoblastic activity at the vertebral endplate surface reaching a peak after 3–4 months, with metabolic activity later progressively decreasing until fusion is achieved, probably after 10–12 months.

This aspect was associated with stability and absence of subsidence at radiological follow-up, confirming that EBM-printed TTCs represent a promising material for reducing hardware complication rates and promoting fusion.

## Figures and Tables

**Figure 1 diagnostics-12-02296-f001:**
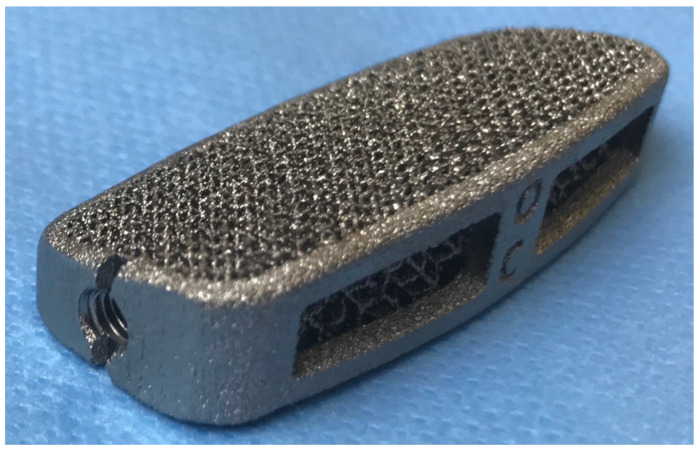
Custom-made trabecular titanium cage for XLIF surgery.

**Figure 2 diagnostics-12-02296-f002:**
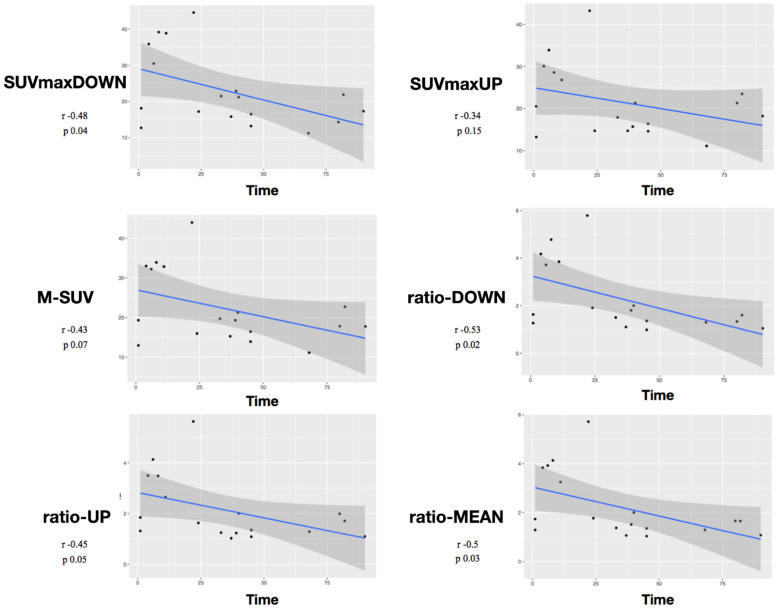
Pearson correlation analysis between all considered variables and time. Scatter plots were drawn to visualize the relationship with a regression line.

**Figure 3 diagnostics-12-02296-f003:**
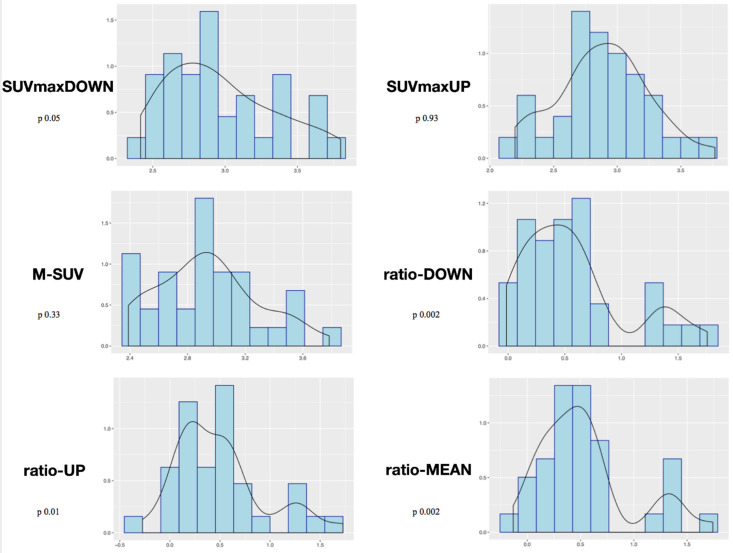
Mixed-effects model analysis with the considered variables and time.

**Figure 4 diagnostics-12-02296-f004:**
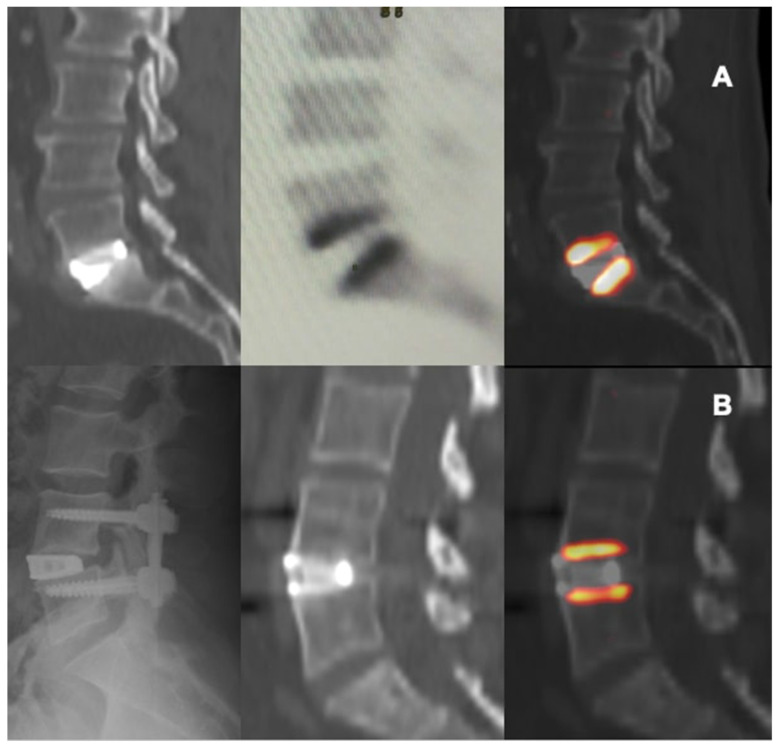
CT and PET images showing metabolic activity on vertebral plates after arthrodesis surgery with trabecular titanium cage. L5-S1 ALIF, 11 weeks (**A**); L4-L5 XLIF, 27 weeks (**B**).

**Figure 5 diagnostics-12-02296-f005:**
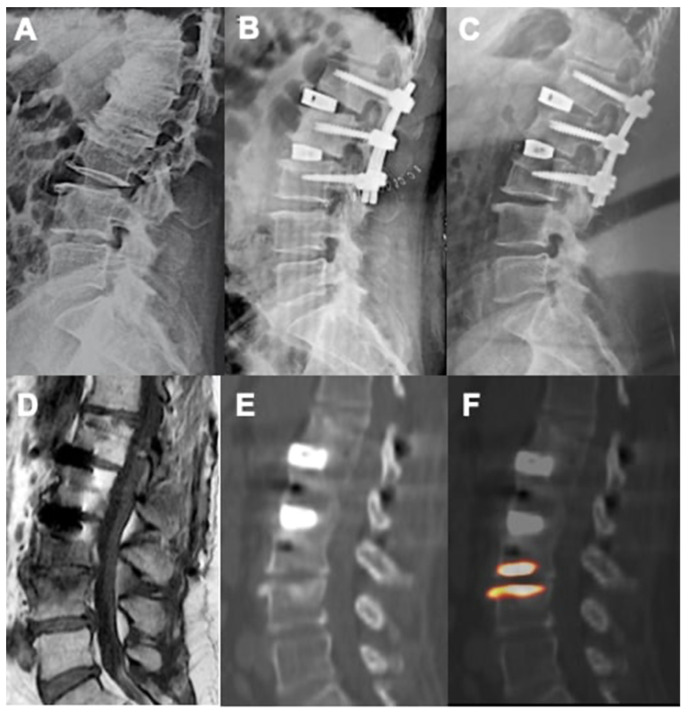
Pre-operative standing X-ray (**A**), immediate postoperative (**B**), and 58 weeks follow-up (**C**) showing good sagittal correction after L1-L2 and L2-L3 XLIF surgery. PET (**F**) and CT (**E**) images showed no activity at the level of the implanted vertebral plates suggesting a proper integration with the bone. Appearance of L3-L4 activity was related to the occurrence of degenerative junctional pathology as also shown by T1-weighted MRI sequences with Modic I signal (**D**).

**Table 1 diagnostics-12-02296-t001:** Descriptive Data.

Patients	15 (7 M, 8 F)
Age (mean/min/max)	61/41/77
Levels	4 L1-L2 (11.5%) 6 L2-L3 (17.1%) 7 L3-L4 (20%) 12 L4-L5 (34.3%) 6 L5-S1 (17.1%)
Type of fusion	XLIF (26), ALIF (9)
Number of levels of surgery	4 Single level (26.7%) 6 Two levels (40%) 2 Three levels (13.3%) 2 Four levels (13.3%) 1 Five levels (6.7%)
Follow-up (mean/min/max)	22.2/13/32

## Data Availability

Not applicable.

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
