# Peer review of "The Efficacy of Trabecular Titanium Cages to Induce Reparative Bone Activity after Lumbar Arthrodesis Studied through the 18f-Naf PET/CT Scan: Observational Clinical In-Vivo Study"

_diagnostics, 2022, doi:10.3390/diagnostics12102296_

Round 1

Reviewer 1 Report

The paper is well written and I have just minor comments and suggestions.

I am missing few references to the general use of [18F]-NaF in bone remodelling diagnosis in the introduction part. 

Please use appropriate indexes, e.g.: 18F and IUPAC recommended notation.

Please explain shortenings at first mention where applicable. 

The quality of Figures 2 and 3 should be improved for better readability. On printed paper they are almost unreadable. X,Y-axes are not clear, missing proper descriptors and units (Fig.2). 

Author Response

Response to reviewer 1

We thank you for the timely and accurate judgment and corrections suggested, here the point-to-point responses:

The paper is well written and I have just minor comments and suggestions.

I am missing few references to the general use of [18F]-NaF in bone remodelling diagnosis in the introduction part. 

R: We have included in the introduction the reasons why PET-CT analysis can be a valuable contribution in bone remodeling diagnosis by inserting the appropriate references

Please use appropriate indexes, e.g.: 18F and IUPAC recommended notation.

R: We rechecked all units of measurement and made suggested corrections (18F and megabequerel (mbq) currently modified)

Please explain shortenings at first mention where applicable. 

R: Corrected

The quality of Figures 2 and 3 should be improved for better readability. On printed paper they are almost unreadable. X,Y-axes are not clear, missing proper descriptors and units (Fig.2). 

R: The proposed images are 300 dpi vector figures that for reasons we do not understand are difficult to read in the pdf, so we have arranged to include them separately in the submission in a dedicated folder.

Reviewer 2 Report

Positron emission tomography (PET) and computer tomography (CT) are undoubtedly very reliable and effective diagnostic medical tools. The paper shows that 18F-NaF PET/CT can serve also to investigate the metabolic-reparative reaction after insertion of titanium trabecular cages. The topic and the level of the manuscript corresponds to the Diagnostics requirements. I would recommend the paper to be published after minor changes.

Comments

1.      Abstract. Line 36. What is EBM?

2.      Introduction. Line 82. I did not quite get reading the article why did the authors decide that 18F-NaF was absorbed by bone hydroxyapatite. Please, clarify this point.

3.      Introduction or Materials and Methods. I suggest introducing a Figure to depict schematically human body and a place for implantation for the broad journal audience.

4.      Conclusions should be re-written to show some essential results obtained in the paper.

Author Response

Positron emission tomography (PET) and computer tomography (CT) are undoubtedly very reliable and effective diagnostic medical tools. The paper shows that 18F-NaF PET/CT can serve also to investigate the metabolic-reparative reaction after insertion of titanium trabecular cages. The topic and the level of the manuscript corresponds to the Diagnostics requirements. I would recommend the paper to be published after minor changes.

We thank you for the timely and accurate judgment and corrections suggested, here the pointo-to-point response:

  1. Abstract. Line 36. What is EBM?

R: corrected. EBM means electron beam melting (EBM)-molded titanium trabecular cages. Added in the text

  1. Introduction. Line 82. I did not quite get reading the article why did the authors decide that 18F-NaF was absorbed by bone hydroxyapatite. Please, clarify this point.

R: After a reevaluation of the literature, we have added to better clarify this point that: Fluorine-18-sodium fluoride (F-NaF) PET/CT is a relatively new, high-resolution bone imaging technique that measures turnover. 18F-NaF PET has previously been shown to correlate with histomorphometric parameters and the osseointegration capacity of different materials, and its use has increased to accurately evaluate images and be aware of normal distribution and major artifacts in imaging.

  1. Introduction or Materials and Methods. I suggest introducing a Figure to depict schematically human body and a place for implantation for the broad journal audience.

R: Figure 4 shows the capture in the image to the best of our ability. This is a relatively new study, and including generic images runs the risk of being unspecific.

  1. Conclusions should be re-written to show some essential results obtained in the paper.

R: We re-written the conclusion as follows: 18F-NaF PET/CT proved to be a reliable tool to investigate metabolic-reparative reaction following the implantation of an interbody fusion cage. Customization for cages and the use of TTCs allows to take advantage of implant stability by maximizing the contact surface area, and also to plan appropriate SL restoration when necessary. 18F-NaF PET/CT demonstrates that TTCs have a good capacity to induce reparative osteoblastic activity at vertebral endplate surface over time reaching a peak after 3-4 months and a later progressive decreasing of metabolic activity until fusion was achieved probably after 10-12 months.